# Humoral and Cellular Responses to a Delayed Fourth SARS-CoV-2 mRNA-Based Vaccine in Weak Responders to 3 Doses Kidney Transplant Recipients

**DOI:** 10.3390/vaccines10091439

**Published:** 2022-09-01

**Authors:** Arnaud Del Bello, Nassim Kamar, Olivier Marion, Jacques Izopet, Florence Abravanel

**Affiliations:** 1Department of Nephrology and Organ Transplantation, Toulouse University Hospital, 31059 Toulouse, France; 2Department of Medicine, Université Paul Sabatier, 31400 Toulouse, France; 3Toulouse Institute for Inflammatory and Infectious Diseases (Infinity), INSERM, 31000 Toulouse, France; 4Department of Virology, Toulouse University Hospital, 31059 Toulouse, France

**Keywords:** solid-organ-transplant recipients, immunodeficiency, kidney transplantation, SARS-CoV-2, vaccine

## Abstract

While kidney transplant recipients (KTRs) represent a high-risk population for severe SARS-CoV-2 infection, almost half of them do not develop adequate levels of antibodies conferring clinical protection despite 3 doses of the mRNA vaccine. In the present study we retrospectively analyzed the humoral and cellular responses resulting from a fourth dose of vaccine administered to KTRs having an anti-SARS-CoV-2 antibody titer below 142 binding antibody unit (BAU)/mL at 3 months post-third-dose. We observed a significant increase in anti-SARS-CoV-2 antibody concentration from 6.1 (Q1 4.3; Q3 12.7) BAU/mL on the day of the fourth dose to 1054.0 (Q1 739.6; Q3 1649.0) BAU/mL one month later (*p* = 0.0007), as well as neutralizing antibody titers (from 0.0 (Q1 0.0; Q3 2.0) to 8 (4; 16) IU/mL, *p* = 0.01) and CD3+ T cell response (from 37.5 (Q1 12.5; Q3 147.5) to 170.0 (Q1 57.5; Q3 510.0) SFUs per 10^6^ PBMCs, *p* = 0.001). Hence, delaying the fourth dose seems to improve vaccine immunogenicity in KTRs, compared with previously reported data obtained in respect of a fourth dose one month after the third dose. Nevertheless, antibody concentrations seem to remain insufficient to confer clinical protection, especially for Omicron, for which breakthrough infections occur even at very high concentrations.

Despite three doses of mRNA vaccine, almost half of solid-organ-transplant (SOT) recipients do not develop adequate levels of anti-SARS-CoV-2 antibodies that confer clinical protection [1]. In several countries, a fourth vaccine dose is recommended in this setting. However, previous reports, including ours, showed poor humoral and cellular responses to a fourth dose given a few weeks after the third dose (range: 4 to 9 weeks) in weak- or non-responders [2,3]. Recently, it has been reported that increasing the interval between two mRNA vaccine doses improved immunogenicity in healthcare workers [4]. Hence, we assessed whether a fourth dose of BNT162b2 mRNA vaccine given at least 4 months after the third dose improves cellular and humoral responses in transplant patients who responded weakly to three BNT162b2 vaccine doses.

A fourth mRNA vaccine dose was proposed for kidney-transplant recipients having an anti-SARS-CoV-2 antibody titer below 142 binding antibody unit (BAU)/mL at 3 months post-third-dose and who declined pre-exposure monoclonal antibody prophylaxis. Twenty patients without a history of COVID-19 infection accepted the fourth dose in our center. In these patients, we measured anti-SARS-CoV-2 spike-binding and neutralizing antibody titers using a Delta variant of the SARS-CoV-2 virus, as well as cellular response (Supporting Information 1) before and one month after the fourth dose. We then compared the results with previously published literature in this population. According to French law (*loi Jardé*), anonymous retrospective studies do not require institutional review board approval.

Twenty kidney-transplant recipients, including a simultaneous pancreas kidney transplant patient (Supporting information 2, Table 1), were given the fourth dose 56 (Quartile 1, (Q1) 22; Quartile 3 (Q3) 112) months post-transplantation, and 5.6 (Q1 4.9; Q3 5.7) months after the third dose. Eighteen patients were given triple immunosuppression. The median anti-SARS-CoV-2 antibody concentration increased from 6.1 (Q1 4.3; Q3 12.7) BAU/mL on the day of the fourth dose to 1054.0 (Q1 739.6; Q3 1649.0) BAU/mL one month later (*p* = 0.0007) (Figure 1A). The median neutralizing antibody titers (Figure 1B) increased (from 0 (Q1 0; Q3 2) IU/mL to 8 (4; 16) IU/mL, *p* = 0.01) and the median CD3+ T cell response (Figure 1C) also increased significantly before and one month after the fourth dose (from 37.5 (Q1 12.5; Q3 147.5) to 170.0 (Q1 57.5; Q3 510.0) SFUs per 10^6^ PBMCs).

Delaying the fourth dose seems to improve the vaccine immunogenicity in SOT recipients. Indeed, binding antibody concentrations at 1 month post-fourth-dose were higher compared to those reported previously in weak responders, i.e., 402 (87–508) BAU/mL^2^ and 145 (27.6–243) BAU/mL^3^ when the last dose was given 65 ± 9 days and 68 (61–74.7) days after the third dose, respectively. Nonetheless, in the absence of randomization or a control group with which to compare the response before and after the fourth dose at different times, we cannot rule out bias in the conclusions of our present study.

However, with the rapid spread of Omicron subvariants, breakthrough infections occur even at very high concentrations (90% < 6967 BAU/mL) in non-immunocompromised healthcare workers [5]. It was demonstrated that the Omicron spike evaded neutralization by antibodies from convalescent patients or individuals vaccinated with the BioNTech-Pfizer vaccine (BNT162b2) with 12- to 44-fold higher efficiency than the spike of the Delta variant [6] These findings suggest that immunocompromised patients with low neutralizing antibodies against the Delta variant would not be sufficiently protected against the Omicron variant. In the present study, only one patient reached this concentration. Neutralizing antibody titers were also low. Other vaccine strategies should be investigated in this population. Nonetheless, Bruminhent et al. assessed the immune response to an inactivated vaccine. They found a weak humoral response but comparable cellular responses in fully vaccinated kidney transplant recipients receiving the inactivated SARS-CoV-2 vaccination compared to immunocompetent individuals [7].

Hence, in this population, pre-exposure monoclonal antibody prophylaxis active against the Omicron variant should be discussed rather than offering a fourth dose, in transplant patients who responded weakly to three doses.

## 1. Supporting Information 1: Virological Methods

### 1.1. Serology Assay

Anti-SARS-CoV-2 spike protein total antibodies were assessed before and after the fourth vaccine using the Wantai semiquantitative microplate ELISA (Wantai SARS-CoV-2 Ab ELISA, Beijing Wantai Biological Pharmacy Enterprise CO., Ltd., Beijing, China). A positive result was defined by a signal-to-cut-off (S/CO) ratio greater than or equal to 1.1. A linear relationship was obtained between the S/CO ratio and SARS-CoV-2 antibody concentration using the first WHO international standard (NIBSC code: 20/136, National Institute for Biological Standards and Control, Potters Bar, Hertfordshire, EN6QG, UK) [8]. Dilutions using PBS plus 7.5% bovine serum albumin as a diluent were set up to analyze samples giving a saturated signal. Results are expressed in binding antibody unit (BAU)/mL. Using this assay, we have previously reported 100% specificity and 100% sensitivity in immunocompetent patients tested at 2 to 14 days post-symptom-onset and at 15 to 45 days post-symptom-onset, suggesting it has the ability to detect low levels of antibodies [9].

### 1.2. Neutralization Assay

Neutralizing antibody titers were assessed using a live virus neutralization assay and a clinical SARS-CoV-2 Delta variant (GISAID: EPI_ISL_10316331) infecting Vero cells (ATCC, CCL-81TM). Briefly, 10^4^ cells were mixed with the virus suspension (100 TCID50) and the tested serum and incubated for 4 days in the wells of 96-well plates. Two-fold serial dilutions of each serum were tested. The plates were then examined to identify the wells showing a cytopathic effect (CPE). The titer was defined as the reciprocal of the highest serum dilution protecting cells from a CPE.

### 1.3. EliSpot Assay

To analyze T-cell responses, enzyme-linked immunospot assays (EliSpot) measuring interferon-γ produced by specific SARS-CoV-2 T-cells were performed, on the day of the fourth dose and 1 month after. PBMCs were thawed and left to recover overnight at 37 °C (2 × 106 cells/mL) in culture medium (RPMI supplemented with glutamine, pyruvate, penicillin, streptomycin and 5% fetal calf serum). ELISpot assays were performed using plates, capture antibodies and detection reagents from the Diaclone kit for detecting IFN-γ. Cell viability was assessed by trypan blue exclusion and 0.4 × 10^6^ viable cells were cultured for 36 h in duplicate wells with antigens in a final volume of 60 μL. The anti-SARS-CoV-2 response was assessed using individual 15-mers 11-aa overlapping peptide pools derived from a peptide scan through SARS-CoV-2 spike glycoprotein (2 pools representing the S1 and S2 domains of the spike protein) (JPT-Peptide-Technologies). Results were expressed as spot-forming unit (SFU)/10^6^ cells. Negative control wells lacked peptides, and positive control wells included CD3/CD28 and CEF pool stimulation. The final peptide concentration was 0.25 μg/mL. Cell function was assessed by polyclonal stimulation in control wells containing 25 × 10^3^ cells stimulated with a mix of anti-CD3 and anti-CD28 antibodies (clones HIT3a and 28.2, respectively, BD Biosciences, 0.5 μg/mL each). The automated Immunospot S6 core reader and software (CTL Europe GmbH) were used to count SFU using SmartCount™ and Autogate™ functions. Spot-forming units of stimulated cells were detected according to the manufacturer’s recommendations. Specific responses were calculated after averaging duplicate wells and subtracting non-specific responses (solvent without peptides).

## 2. Supporting Information 2: Clinical Informations of Included Patients

**Table 1 vaccines-10-01439-t001:** Patients’ characteristics.

	*N* = 20
Gender (M/F)	11/9
Age (years, mean ± SD)	52 ± 14
History of rejection in the year preceding vaccination, n	0
Time between vaccine and transplantation (months, IQR 1; 3)	56 (22; 112)
No induction therapy, n (%)	8 (40)
Induction therapy, n (%)	12 (60)
Anti-IL2 receptor blockers	5 (42)
Polyclonal antibodies	7 (58)
Type of immunosuppressive regimen *, n (%)	
Calcineurin-inhibitors	17 (85)
Tacrolimus	16 (80)
Ciclosporin A	1 (5)
Mycophenolic acid	13 (65)
mTOR inhibitors	7 (35)
Steroids	18 (90)
Belatacept	3 (15)
Neutrophil count before vaccination (/mm^3^, mean ± SD)	5622 ± 2842
Lymphocyte count before vaccination (/mm^3^, mean ± SD)	1571 ± 764
CD4+ T-cell count before vaccination (/mm^3^, mean ± SD)	546 ± 311
CD8+ T-cell count before vaccination (/mm^3^, mean ± SD)	451 ± 308
CD19+ T-cell count before vaccination (/mm^3^, mean ± SD)	91 ± 85
NK cell count before vaccination (/mm^3^, mean ± SD)	253 ± 232
eGFR before vaccination (mL/min/1.73 m^2^)	51 ± 19
Neutrophil count before 4th dose (/mm^3^, mean ± SD)	6125 ± 2978
Lymphocyte count before 4th dose (/mm^3^, mean ± SD)	1061 ± 805
eGFR before 4th dose (mL/min/1.73 m^2^)	46 ± 23

* Target tacrolimus C0: 5–7 ng/mL, target ciclosporin A C2: 500–1000 ng/mL, everolimus C0: 3–8 ng/mL. Mycophenolic acid was taken at 360 mg b.i.d, prednisone at 5 mg/d, and belatacept at 5 mg/kg monthly.

## 3. Conclusions

In this population, pre-exposure monoclonal antibody prophylaxis active against the Omicron variant should be used rather than offering a fourth dose, in transplant patients who responded weakly to three doses.

## Figures and Tables

**Figure 1 vaccines-10-01439-f001:**
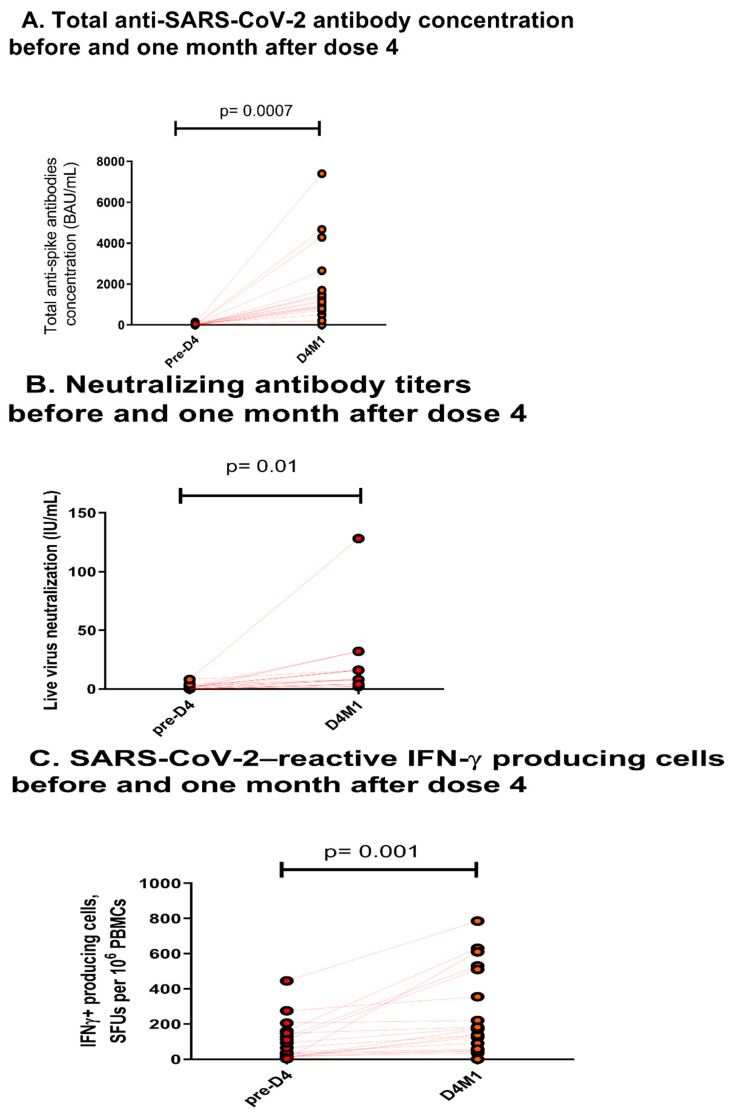
(**A**). Total anti–SARS-CoV-2 antibody concentration before and one month after the fourth dose. Data are expressed in binding antibody unit (BAU)/mL. *p* values were derived from Student’s paired *t*-test. The solid lines indicate the medians. (**B**). Neutralizing antibody titers before and one month after the fourth dose. Data are expressed as the reciprocal of the highest serum dilution protecting cells from a cytopathic effect. The solid lines indicate the medians. (**C**). The numbers of cells reactive to overlapping peptide pools spanning SARS-CoV-2 structural protein S (pools S1 and S2). Data are expressed by spot-forming units (SFUs) per 10^6^ peripheral blood mononuclear cells (PBMCs). The solid lines indicate the medians.

## Data Availability

Upon reasonable request.

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
