# Peer review of "Humoral and Cellular Responses to a Delayed Fourth SARS-CoV-2 mRNA-Based Vaccine in Weak Responders to 3 Doses Kidney Transplant Recipients"

_vaccines, 2022, doi:10.3390/vaccines10091439_

Round 1

Reviewer 1 Report

Major:

1. Could the authors indicate the degree of immunosuppression, i.e. the doses of immunosuppressive regimens received by the subjects?

2. For paired analyses, could the authors indicate pairing on the figures? (i.e before and after vaccination)

3. It is not certain that the discussion on Omicron is appropriate, esp. with data collected by the authors themselves. Knowledge landscape is evolving dynamically and correlates of protection are not yet well defined. However, indeed the authors should acknowledge the limitations of the study with regards to currently circulating variant (Omicron). I would suggest replacing this part with merely an acknowledgement of the limitations (rather than direct recommendation of monoclonal antibodies), but it is up to the Editors to decide.

Minor

1. Please re-formulate lines 42-43. The point was understandable, but not grammatically correct. "...who responded weakly to..." might be more appropriate.

2. "Pre-vaccination" and "D4M1" to indicate before vaccination and "1 month after 4th dose" might be appropriate.

In general: please consult a native English speaker for extensive English editing.

Author Response

Major:

  1. Could the authors indicate the degree of immunosuppression, i.e. the doses of immunosuppressive regimens received by the subjects?

As requested, targeted doses of IS drugs were mentionned in the supporting Table 2.

  1. For paired analyses, could the authors indicate pairing on the figures? (i.e before and after vaccination)

As requested, figures were modified to indicate pairing.

  1. It is not certain that the discussion on Omicron is appropriate, esp. with data collected by the authors themselves. Knowledge landscape is evolving dynamically and correlates of protection are not yet well defined. However, indeed the authors should acknowledge the limitations of the study with regards to currently circulating variant (Omicron). I would suggest replacing this part with merely an acknowledgement of the limitations (rather than direct recommendation of monoclonal antibodies), but it is up to the Editors to decide.

We agree with reviewer’s comment. Nevertheless, it was demonstrated that the Omicron spike evaded neutralization by antibodies from convalescent patients or individuals vaccinated with the BioNTech-Pfizer vaccine (BNT162b2) with 12- to 44-fold higher efficiency than the spike of the Delta variant (Hoffmann Cell 2022). This finding suggested that in our immunocompromised patients with low neutralizing antibodies against the delta variant would not be sufficiently protected against the Omicron variant. This study is now added in the revised manuscript.

Minor

  1. Please re-formulate lines 42-43. The point was understandable, but not grammatically correct. "...who responded weakly to..." might be more appropriate.

This was corrected

  1. "Pre-vaccination" and "D4M1" to indicate before vaccination and "1 month after 4th dose" might be appropriate.

As requested this was corrected

In general: please consult a native English speaker for extensive English editing.

            As requested, this was done

Reviewer 2 Report

1. Briefly summarize the content of the manuscript; Authors have studied the delayed humoral and cellular response to a fourth dose of m RNA vaccine given at least 3 months after a third m RNA vaccine in kidney transplant recipients. They observed that both the levels of anti-SARS-COV2 antibodies and neutralizing antibodies were remarkably raised compared to few weeks after third dose of vaccine. However the level of titres were not as high as expected to confer immunogenicity against new COVID-19 infections, specifically Omicron variants

2. The manuscript’s strengths of the manuscript is the attempt to study delayed responses of fourth dose of fourth vaccine as previous attempts studied only responses within few weeks after vaccination. The weakness is the relatively small sample size that included only 20 participants. Authors have not described the characteristics of the study participants in terms of socio-demographic and clinical profiles; hence, it is difficult to comments on antibody responses being affected by confounders. The study site is not described, presumably a single centre, and with small number of participants, these findings cannot be generalized.

 3. Major recommendations for the improvement of the manuscript; Authors have concluded that the level of antibodies increase is not enough to confer immunity against Omicron variants breakthrough infection, however they have used health care workers as a reference point. This cannot be a conclusion for this study, as the study never studied omicron breakthrough infections. It is however still a worthwhile discussion point.

4. Minor recommendations for the improvement of the manuscript: Several typing errors needs to be addressed. These are indicated below;-

·       Lines 53-55: “Twenty kidney-transplant recipients, including a simultaneous pancreas kidney 53 transplant patient (Supporting information 2), were given the fourth dose 56 (Q1 22; Q3112) months post transplantation and 5.6 (Q1 4.9; Q3 5.7) months after the third dose: Did all the participants undergo transplantation 56 months before fourth m RNA vaccine? Add details to indicate what Q1, Q2, Q3 and Q4 represent

·       Line 34-35:  Correct the phrase from solid-organ-transplant recipients (SOT) into solid-organ-transplant (SOT) recipients

·       Lines 66-68: The statement “Nevertheless, antibodies concentrations remain insufficient to confer a clinical protection, especially for Omicron against for whom breakthrough infections occurred even at very high concentration (90% < 6967 BAU/ml) in health care workers f5”. Authors should correct or provide explanation as to whether health care workers were part of this study. Did the author study occurrence of Omicron breakthrough infection among the 20 study participants? What does “f5” represent?

Author Response

  1. Major recommendations for the improvement of the manuscript; Authors have concluded that the level of antibodies increase is not enough to confer immunity against Omicron variants breakthrough infection, however they have used health care workers as a reference point. This cannot be a conclusion for this study, as the study never studied omicron breakthrough infections. It is however still a worthwhile discussion point.

As proposed a correction was made in the conclusion.

4. Minor recommendations for the improvement of the manuscript: Several typing errors needs to be addressed. These are indicated below;-

  • Lines 53-55: “Twenty kidney-transplant recipients, including a simultaneous pancreas kidney 53 transplant patient (Supporting information 2), were given the fourth dose 56 (Q1 22; Q3112) months post transplantation and 5.6 (Q1 4.9; Q3 5.7) months after the third dose: Did all the participants undergo transplantation 56 months before fourth m RNA vaccine? Add details to indicate what Q1, Q2, Q3 and Q4 represent

Q1 and Q3 represent Quartile 1 and 3 respectively. The median time between transplantation and the fourth dose was 56 months with an interquartile range about : quartile 1 : 22 months, and quartile 3 : 112 months. Clarifications about abbreviations were added in the manuscript.

  • Line 34-35:  Correct the phrase from solid-organ-transplant recipients (SOT) into solid-organ-transplant (SOT) recipients

This was done

  • Lines 66-68: The statement “Nevertheless, antibodies concentrations remain insufficient to confer a clinical protection, especially for Omicron against for whom breakthrough infections occurred even at very high concentration (90% < 6967 BAU/ml) in health care workers f5”. Authors should correct or provide explanation as to whether health care workers were part of this study. Did the author study occurrence of Omicron breakthrough infection among the 20 study participants? What does “f5” represent?

In fact f5 is an error, and represent the reference 5 (Dimeglio, JI 2022). This was corrected.

As proposed above, discussion about Omicron spike evaded neutralization by antibodies vaccinated with the BioNTech-Pfizer vaccine was added.

Reviewer 3 Report

This is an important addition to the COVID literature. The overall conclusion is sound in that it may be advisable to use monoclonal antibodies as pre-exposure prophylaxis in kidney transplant recipients who may be severely immune-compromised. However, there are two additional aspects that could significantly enhance the manuscript.

·       First, is there any data in this population with other vaccine platforms and/or heterologous boosts? Or data with infected patients? For example, it would be good to understand the context of whether or not a heterologous dosing regimen (i.e. mRNA platform followed by Ad-vectored or recombinant protein based vaccine) may elicit higher antibody titers.

·       Line 67: The data may not support the statement around Omicron. This would be better supported if an Omicron variant was used in the ELISA and neutralizing antibody assays. While the use of Omicron based assays will almost assuredly NOT change the conclusions, that caveat should be noted.

Minor comments:

·       Line 44: “was proposed to kidney-transplant recipients having…”

·       Line 48-50: Suggest rewording as “In these patients, we measured anti-SARS-CoV-2 spike-binding and neutralizing antibody titers using a delta variant of SARS-CoV-2 as well as cellular responses (Supporting Information 1) before…”.

·       Line 59-61: Suggest rewording in a similar manner as that of the binding antibody titers in lines 56-68.

·       Line 65-66: I’m not sure if there was a typo in this section, “… when the last dose was given 65  9 days and 68 (61-74.7) days…”

Author Response

First, is there any data in this population with other vaccine platforms and/or heterologous boosts? Or data with infected patients? For example, it would be good to understand the context of whether or not a heterologous dosing regimen (i.e. mRNA platform followed by Ad-vectored or recombinant protein based vaccine) may elicit higher antibody titers.

Other vaccine strategies were poorly investigated in this population until now. Nonetheless, Bruminhent et al. have assessed the immune response to an inactivated vaccine. They found a weak humoral response but comparable cellular responses in fully vaccinated kidney transplant recipients receiving the inactivated SARS-CoV-2 vaccination compared to immunocompetent individuals (Bruminhent,  Am. J. Transplant. 2021). This information was added in the manuscript.

  • Line 67: The data may not support the statement around Omicron. This would be better supported if an Omicron variant was used in the ELISA and neutralizing antibody assays. While the use of Omicron based assays will almost assuredly NOT change the conclusions, that caveat should be noted.

 We recognized that the use of the omicron for the neutralizing and the Elisa antibodies assays could be interesting.  However, very few validated commercial Elisa used the omicron variant’s antigens. Our study was conducted in December 2021. At that time, the omicron variant has not emerged yet in France and the neutralizing assay used the delta variant. However, Hoffmann et al have demonstrated that the Omicron spike evaded neutralization by antibodies from convalescent patients or individuals vaccinated with the BioNTech-Pfizer vaccine (BNT162b2) with 12- to 44-fold higher efficiency than the spike of the Delta variant (Hoffmann Cell 2022), supporting our conclusion.

Minor comments:

  • Line 44: “was proposed to kidney-transplant recipientshaving…”

This was corrected

  • Line 48-50: Suggest rewording as “In these patients, we measured anti-SARS-CoV-2 spike-binding and neutralizing antibody titers using a delta variant of SARS-CoV-2 as well as cellular responses (Supporting Information 1) before…”.

As proposed, this was changed

  • Line 59-61: Suggest rewording in a similar manner as that of the binding antibody titers in lines 56-68.

As proposed this was changed

  • Line 65-66: I’m not sure if there was a typo in this section, “… when the last dose was given 65  9 days and 68 (61-74.7) days…”

Typo error was corrected.

Reviewer 4 Report

This is an interesting report on the "hot topic" of immune response induced by vaccines generally and more specific by Covid vaccines in transplanted patients. I have a concern

Authors concluded that "delaying the fourth dose seems to improve the vaccine immunogenicity in KTR " by comparing immune results before and one month after a "delayed" forth dose. 

To reach in such a results should better compare immune results before and one month after a "delayed" forth dose with those of before and one month after a "non delayed" forth dose.

Please clarify on this  

Author Response

This is an interesting report on the "hot topic" of immune response induced by vaccines generally and more specific by Covid vaccines in transplanted patients. I have a concern

Authors concluded that "delaying the fourth dose seems to improve the vaccine immunogenicity in KTR " by comparing immune results before and one month after a "delayed" forth dose. 

To reach in such a results should better compare immune results before and one month after a "delayed" forth dose with those of before and one month after a "non delayed" forth dose.

Please clarify on this

We thank the reviewer for this comment. 

The present study is a retrospective analysis of solid organ transplant patients who responded weakly to 3 doses of BNT16B2 vaccine (and who declined pre-exposure monoclonal antibodies). Hence, the design of the study prevents us from randomization or control group. However, we compared results of the fourth dose, in patients that received the 4th dose early after the 3rd dose (including a previous work by our group, with highly comparable patients).

This information was added in the text. 

Round 2

Reviewer 4 Report

I have no further comments